# Empowering Women in Finance through Developing Girls’ Financial Literacy Skills in the United States

**DOI:** 10.3390/bs11120176

**Published:** 2021-12-10

**Authors:** Chong Myung Park, Aidan D. Kraus, Yanling Dai, Crystal Fantry, Turner Block, Betsy Kelder, Kimberly A. S. Howard, V. Scott H. Solberg

**Affiliations:** 1Wheelock College of Education and Human Development, Boston University, Boston, MA 02215, USA; ekkraus@bu.edu (A.D.K.); yldai@bu.edu (Y.D.); tblock@bu.edu (T.B.); khoward@bu.edu (K.A.S.H.); ssolberg@bu.edu (V.S.H.S.); 2Invest in Girls, New York, NY 10168, USA; cfantry@investgirls.org (C.F.); bkelder@investgirls.org (B.K.)

**Keywords:** financial literacy, financial education, career development, female leaders

## Abstract

This study examines the effectiveness of a financial literacy program, Invest in Girls (IIG), in promoting financial capability among high school girls. Using a quasi-experimental separate-samples pretest-posttest design and a longitudinal qualitative study, the study aims to assess the program efficacy and investigate the perspectives of the female students on its impact on their knowledge, behavior, and future goals and aspirations. The results indicated that the participants had significantly higher confidence for engaging in financial literacy after the program. The findings from the longitudinal study also suggested that that the program was influencing the students in positive ways, increasing their financial capability and leading them to consider wide occupational pathways available in finance. Given the lack of female leaders in the world of finance, the IIG program aims to address gender disparity in financial knowledge and highlight the importance of building financial literacy skills among girls.

## 1. Introduction

Twenty-two percent of fifteen-year-olds in the United States can, at best, recognize the purpose of an invoice, and only ten percent can respond appropriately to a financial scam email message [1] According to the PISA financial literacy assessment that assessed the financial literacy skills of 1486 US students and compared across partner countries, the United States ranked sixth among the fifteen participating nations. In Canada, where students go through similar stages of academic learning from elementary to postsecondary education and experience diversity in schools, which is more focused in urban settings, thirteen percent of students were below Proficiency Level 2 when it came to being able to recognize the purpose of an everyday financial document, and twenty-two percent were able to respond appropriately to a scam message (Proficiency Level 5). The assessment also found that socio-economically disadvantaged students in the United States were more than twice as likely as advantaged students to be unable to recognize the purpose of an invoice (Level 2). The PISA assessment is meaningful in that it examined the knowledge and skills of 15-year-olds who were starting to make decisions on how to spend their pocket money, set spending priorities, and be aware of potential scam messages. The assessment also alerts and asks us to reconsider how we, as educators, are preparing youth for their transition to adulthood and whether we are giving enough information and opportunities to youth—female students, in particular—to help them consider occupations within the financial industry. The PISA financial literacy assessment informed us that more boys than girls were found in the top performers and provided a policy implication that gender inequity needs to be addressed in providing learning opportunities [1].

Understanding how financial literacy is taught in school in the United States is complicated as each state has its own requirements that must be fulfilled before an offer is made regarding an optional course, which may entail the teaching personal finance within another course like mathematics or economics, or teaching the subject as a standalone course. Invest in Girls is a financial literacy program that works with schools and community organizations to provide financial education to young girls with an aim to ultimately address the gender inequity issue in the financial industry. As part of this effort, this paper aims to examine the effectiveness of the IIG program in promoting financial literacy among high school girls and its impact on their knowledge, behavior, and decisions for the future. This paper provides an overview of how financial literacy and education are defined in this study and explores the existing literature addressing the gender gap in financial literacy. The study introduces Individualized Learning Plan (ILP) as a conceptual model that promotes financial literacy as part of career planning and management strategies. We discuss how state policies in the United States have been responding to the needs of financial literacy education, which helped shape the IIG curriculum, and how the ILP has been playing the role of promoting financial literacy in different states. Invest in Girls is compared with the national standards to see its alignment. The effectiveness of the IIG program is evaluated using a separate-sample pretest-posttest design that provides both quantitative and qualitative evidence. The program’s impact on girls is examined through a longitudinal qualitative study.

### 1.1. Definitions of Financial Literacy

The existing literature provides various definitions of financial literacy, from narrow ones that focus on one’s knowledge, to broad ones that extend to our social responsibility. This paper describes these various definitions in order to highlight the role financial education plays in addressing different societal issues, including the gender wage and career advancement gaps.

The JumpStart Coalition, a non-profit organization that created the national standards in K-12 personal finance education, identifies two key elements across different definitions of financial literacy. First, financial literacy refers to one’s knowledge of personal finance, and second, it concerns one’s ability to use the information and resources that are key to achieving and maintaining his/her financial wellbeing [2]. With the use of obtained knowledge and information, one can understand the news and economic information and make informed decisions about spending, debt, retirement, and wealth accumulation [3]. This is consistent with how financial capability is defined by the FINRA foundation [4]; that it is not a simple concept but encompasses multiple aspects of behavior, factors, and skill sets used in resource management and financial decisions. The worldwide pandemic that affected us all also makes us consider an understanding of the changing economic circumstances of individuals when defining what consists of financial literacy. Financial literacy can involve the adaptability that enables individuals to respond to the changes effectively and make appropriate financial decisions [2]. The OECD provides a comprehensive definition of financial literacy that extends to our social responsibility, “knowledge and understanding of financial concepts and risks, and the skills, motivation, and confidence to apply such knowledge and understanding in order to make effective decisions across a range of financial contexts, to improve the financial well-being of individuals and society, and to enable participation in economic life” [5].

While often used interchangeably, financial education has an emphasis on the teaching of financial content, while financial literacy focuses on one’s ability to apply learned knowledge that leads to new behavior. In other words, financial education emphasizes the “process” of building knowledge and the understanding of financial concepts, and financial literacy focuses on the action of “applying” and “improving” one’s present and future financial well-being [6]. Based on these definitions, it can be concluded that teaching financial literacy is not only the activity of delivering information or knowledge but in the broader sense empowering and motivating people to think about their behavior, possibly changing them to make wiser financial decisions, and improving their relevant skills constantly for their lifelong well-being [2,7]. Given the broad impact of financial literacy on youth, from their everyday lives to lifelong well-being, financial education and literacy can serve as a channel for addressing inequities existing in our society, especially the gender wage and career advancement gaps.

### 1.2. Gaps in Financial Literacy and the Positive Impact of Financial Education

We live in a society where women represent only a small portion of executives and board members. In 2019, only 22 percent of leadership positions at US financial-services firms were filled by women [8]. Although the number is anticipated to grow, the proportion of female leaders in the financial industry will still be well below parity at 31% in 2030. The lack of female leaders provides fewer role models for girls, discouraging them from seeing such industries as viable careers that they can be successful in. Female students see themselves as less “interested” in financial matters and report lower self-knowledge than their counterparts [9]. As they transition to adulthood, the trend of low self-esteem seems to continue as the knowledge gap widens. It was found that American women were less likely than American men to give correct answers to what they called “the Big Three Questions” on interest rates, inflation, and risk diversification regardless of their socio-economic, cultural, and institutional backgrounds [10].

While efforts have been made to promote financial literacy through different financial education programs, financial literacy, in fact, was the topic least likely to be addressed (5.6%), according to a meta-analysis of the educational interventions that prepared adolescents for adulthood [11]. Within this limited spectrum, only a few intervention studies have addressed the gender disparity [7]. After an intervention on several financial concepts, it was reported that the overall scores for both genders increased, however the confidence scores of male teens were higher than female teens before and after the intervention [12]. The confidence scores among female teens increased to a more considerable degree, partly as they were less familiar with some of the financial concepts before the intervention [12]. Rothwell and Wu (2019) found that the individuals who had taken a financial education course had higher subjective knowledge and self-efficacy scores for both genders and across ages. However, they also found that higher objective knowledge scores were partially driven by higher scores of male participants, concluding that a gender gap existed in objective financial knowledge.

The objective financial knowledge is deemed critical as it triggers the positive cycle of developing financial capability, especially when taught to adolescents [13]. The existing literature indicates three major areas of financial capability youth can develop through financial education, including behavior, knowledge, and autonomy, with the majority being focused on behavior. Kaiser and Menkhoff [14] conducted a meta-analysis of 126 intervention studies and found that financial education had a positive, measurable impact on financial behavior. In examining the effect of the states’ financial education policies on students’ credit behavior, Urban and her colleagues [15] reported that their credit behavior improved after the states implemented financial education requirements. Friedline and West [16] examined the specific financial behavior of more than 6000 US Millennials and found that those financially “capable”—who received financial education and were financially included with a savings account—were more likely to afford unexpected expenses and save for emergencies and less likely to use alternative financial services and carry burdensome debt. The debt effects of financial education have been examined by several researchers, such as improved repayment behavior and decreased reliance on non-student debt [15,16,17]. A program in Chile also reported that financial education decreased the total debt of the participants in 3 and 6 months after the training [18]. In her 3-year longitudinal study in Ghana, Clark [19] found increased savings among the participating girls after learning about savings in the first year, followed by an opportunity to save $15 per year in the next two years. Walstad and his colleagues [20] also found that financial education could change students’ financial aid application completion rate. The positive impact of financial education on behavior can be seen among elementary school students as well. Working with 110 elementary schools in Chicago, Hegadorn and his colleagues (2016) found wiser spending habits among the young students after the intervention.

Financial education programs have a positive impact on both financial behavior and knowledge [20,21,22,23]. Financial education programs improve the financial knowledge of high school students and raise their interest in learning more about finance [9]. Although not discussed in the study, it is possible that the students could explore occupational opportunities aligned with the financial industry by developing knowledge, interest, and relevant skills, potentially leading them to enter such career pathways in the future.

In addition to the behavior and knowledge that can be developed through financial education, a financially literate person can also manage their funds independently and make decisions, referred to as financial autonomy. A pretest-posttest study of a financial education program in India that provided workshops to 300 homemakers found that the participants’ financial autonomy improved after the intervention in three areas: the ability to think before acting, confidence in one’s choice, and self-control [24]. Although focused on the adult population, this study is meaningful in that it addresses how to develop financial autonomy among female populations who might have been oppressed in exercising their ability.

An increasing number of studies focus on financial behavior, knowledge, and autonomy. However, there is still room for more studies investigating how financial education influences female students, their transition to adulthood, and the occupational opportunities that can open up with increased financial literacy. Financial education programs have the potential to reduce the gender gap and increase the number of women representing the finance field and our society.

As part of an effort to understand how financial literacy can help reduce the gender gap issue and increase the number of female leaders in finance, this paper uses Individualized Learning Plan as a conceptual model that promotes financial literacy skills within the frame of career development. ILPs have shown promise in preparing youth for the transition to adulthood and helping them become “future-ready” by identifying occupational goals and building skills aligned to the goals that are transferable to a wide range of occupations [25,26]. Quality ILPs consist of: (1) a document that lists students’ ILP activities; (2) the process of youth identifying future goals and recording/updating their personalized pathway aligned to these goals [27]. There are three main areas of the career development process that an ILP program can be organized around: self-exploration skills, career exploration skills, and career planning and management skills. As a strategy for developing career management skills, ILP introduces financial literacy that increases college and career readiness for all students to become “future-ready.” As such, the expected outcomes of the IIG program are increased career management skills and readiness for the future.

### 1.3. State Policies on Financial Education

The effectiveness of many financial education interventions encouraged many US states to adopt financial literacy requirements and urged the creation of national standards that provide guidance to different programs for consistency. Nearly half of the states included personal finance courses as of 2015 [15]. While more schools are requiring their students to complete financial education before graduation, some are more or less rigorous by adding financial education as an optional course or providing it with a minimal standard. In 2016, seventeen states required a high school personal finance course, and only five states required a semester course focusing on personal finance [20]. More recent studies indicate that the type of mandate varies by state. Some states require schools to offer a course and leave it up to the students to take it, such as in Nebraska and New Mexico [28].

Some states are doing a remarkable job at helping young people build their financial literacy skills. Georgia is the first state that approved a statewide financial education mandate in 2004. The state’s policy includes student performance testing and standardized content across schools by providing funding for two experts to provide training to the teachers who will be teaching the course [15]. Personal financial literacy in Oklahoma starts from grade seven to twelve with a strong monitoring system that cumulatively records related coursework on high school transcripts [29,30]. Efforts have been made in Wisconsin to promote financial literacy across the state. Forty-four percent of school districts required a course in personal financial literacy in 2013, which increased to 70% in 2018 [31]. In 2020, Wisconsin added a strand on “financial mindset” that addressed the mental habits of students when making financial decisions [32].

### 1.4. The Role of Individualized Learning Plans in Promoting Financial Literacy

Individualized Learning Plan (ILP) is a college and career readiness initiative that has been adopted by thirty-three states as a mandate and in use by ten more states without a mandate in the United States [33]. Many states have included financial literacy as one of the core components of state-wide ILP implementation [27], which enables students to develop financial literacy skills as part of their career planning and management skills. Coupled with the state policies on financial education, more students now have access to financial education. Below describes how financial education is incorporated into the state ILP mandates using state examples.

ILP is mandated in Arizona. The Education and Career Action Plan asks students to create financial assistance plans as part of entering and updating their postsecondary education goals [34]. The District of Columbia mandated ILP for all public high schools, and Florida only ordered middle schools. Both states require instructions on postsecondary financial aid, how to apply for financial aid, and other resources [35,36]. Alaska has a mandate in guidance, as opposed to statute, and includes financial education as part of Academic Development, where schools are recommended to include a component of financial aid and scholarship information [37]. In Colorado, financial literacy is an expectation for high school graduation, with a focus on financial aid topics and vocabulary and an understanding of college financial options [38]. Students are expected to apply the knowledge to their postsecondary and career planning. The current policy efforts by many states concerning financial literacy are important in that they are helping students compare relative costs of a 2-year and a 4-year college, public and private institutions, and in-state and out-of-state tuition fees and weigh whether their postsecondary education and career plans are likely to result in high wages [27].

While efforts have been made to promote financial literacy among our youth by state leaders and researchers, it is clear that gender inequity in financial literacy is an issue that has not been addressed much across the systems, at the policy, district, and school levels. Given the role that financial literacy plays in raising women’s financial wealth [39], autonomy [24], and behavior [15,16,17], this study adds values to the existing literature by examining the effectiveness of a financial literacy program on female students and how it impacts their skills, transition to adulthood, and decision for postsecondary education and career goals.

## 2. Invest in Girls

Invest in Girls (IIG) is a program of the Council for Economic Education [40] that aims to change the trend of a small portion of female executives and board members in our society. As IIG attempts to empower female students by providing a quality financial literacy program, we were asked to evaluate whether the IIG curriculum is supporting girls’ financial literacy development. IIG uses a holistic approach where students learn financial concepts with real-world news and information (Workshops), develop personal finance skills (Workshops), and build knowledge and skills enough to enter the pathways to the financial industry (Role Models & Industry Trips). Three workshop modules are offered to high school female students: CFO workshops focus on budgeting and personal finance; CIO workshops delve into investing money; and CEO workshops focus on college debt, taxes, insurance, interviewing, and philanthropic giving. Each module consists of four workshops:CFO: Becoming the CFO of your life:
Workshop 1: Students are instructed how to become aware of their “money personality”, as well as how to create S.M.A.R.T. goals;Workshop 2: This workshop addresses money management, including how to create a savings and spending plan, how to use “delayed gratification” when pursuing financial goals, and the difference between fixed and variable expenses;Workshop 3: This workshop discusses the importance of saving money as well as the ability to distinguish between different financial accounts. Students are also taught the difference between simple and compound interest;Workshop 4: Students are introduced to credit concepts and credit cards, including the differences between a debit card and a credit card, the benefits and problems of using a credit card, and how to use a credit card responsibly.
CIO: Becoming the CIO of your life:
Workshop 1: This workshop introduces students to basic investment concepts and the stock market;Workshop 2: This workshop discusses the importance of diversification when investing;Workshop 3: This workshop discusses the importance of retirement savings and the difference between a 401k plan, IRA, and Roth IRA;Workshop 4: This workshop discusses different careers in finance. Students learn about the different qualifications, education, and certifications required in order to be able to work in various finance careers.
CEO: Becoming the CEO of your life:
Workshop 1: In this workshop, students learn about how to pay for college using student loans, as well as the different options available to repay them;Workshop 2: This workshop discusses income tax and introduces students to different life situations that may require insurance;Workshop 3: This workshop discusses what makes a strong resume, as well as how to decide between multiple job offers and the importance of salary negotiations;Workshop 4: This workshop introduces students to philanthropy and the various reasons or ways someone can be philanthropic.

In addition to the workshops on different topics, students have an opportunity of going on industry trips that help them witness how it is like to be working in the financial industry. The role model component of the IIG program provides guidance that is tailored to the needs of individual students and their career aspirations by connecting industry female leaders with youth. Bucher-Koenen and her colleagues [10] found that women had difficulties obtaining professional financial advice. This mentoring component helps the female students not only develop field knowledge but also envision themselves as female leaders in finance.

### Curriculum Comparison with the National Standards

The Jumpstart Coalition, partnering with business, government, academia, schools, and other sectors, developed a national standard that delineates financial knowledge and skills youth should acquire from kindergarten through to twelfth grade. It is designed to provide guidance to school and extra-curricular programs and promote consistency, not a uniform model [3]. Most importantly, the standards support the lifetime financial well-being of students, which is consistent with the definitions of financial literacy/education created by the US government and the OECD. The standards have been updated since its first publication in 1998, and the information used to compare with the IIG curriculum is from their fourth iteration effort. The Jumpstart Coalition and the Council for Economic Education, which provides another set of financial literacy standards, are working together to create a single set of national standards that would provide more consistency in financial education (to be released in fall 2021). The national standards used in this study involve six categories of financial competency, and each category includes 3–8 standards.

In comparing the national standards to the IIG’s curriculum, it was found that out of twenty-six standards across six categories, the IIG curriculum was aligned with twenty-one standards. The other five standards were deemed as either having not been met, or as requiring further information for more accurate comparisons. For each category, the standards that were aligned with the IIG curriculum are described, and the standards that were deemed misaligned are further explored with suggestions on what areas of the existing IIG curriculum might be a place to include the missing information (see Implication for Practice for more detailed suggestions). A summary matrix table can be found in Appendix A.

In the ‘Spending and Saving’ category, ‘Develop a plan for spending and saving’ (Standard 1) overlapped with the contents covered in the ‘CFO workshop 2: Money Management’. ‘Develop a system for keeping and using financial records’ (Standard 2) was aligned with the information covered in the ‘CFO Workshop 2: Money Management’. ‘Describe how to use different payment methods’ (Standard 3) overlapped with the content delivered during the ‘CFO Workshop 4: Introduction to Credit Cards’. ‘Apply consumer skills to spending and saving decisions’ (Standard 4) was aligned with the information covered in the ‘CFO Workshop 2: Money Management’.

In ‘Credit and Debit’, ‘Analyze the costs and benefits of various types of credit’ (Standard 1) overlapped with the information taught in the ‘CFO Workshop 4: Introduction to Credit Cards’. ‘Summarize a borrower’s rights and responsibilities related to credit reports’ (Standard 2) was aligned with the ‘CFO Workshop 4: Introduction to Credit Cards’. ‘Apply strategies to avoid or correct debt management problems’ (Standard 3) was also covered during the ‘CFO Workshop 4: Introduction to Credit Cards’. Lastly, ‘Summarize major consumer credit laws’ (Standard 4) was found to not have been covered by the IIG curriculum, however this was deemed to be appropriate material for inclusion in the ‘CFO Workshop 4: Introduction to Credit Cards’.

‘Employment and Income’ was a category that showed alignment across all standards. ‘Explore job and career options’ (Standard 1) was covered during the ‘CIO Workshop 4: Careers in Finance’. ‘Compare sources of personal income and compensation’ (Standard 2) was aligned with the ‘CFO Workshop 2: Money Management’. ‘Analyze factors that affect net income’ (Standard 3) was consistent with the information covered in the ‘CEO Workshop 2: Income Tax & Insurance’.

In the ‘Investing’ category, ‘Explain how investing may build wealth and help meet financial goals’ (Standard 1) overlapped with the ‘CIO Workshop 1: Basic Investment Concepts’. ‘Evaluate investment alternatives’ (Standard 2) was aligned with the ‘CIO Workshop 2: Investment Diversification’. ‘Demonstrate how to buy and sell investments’ was covered during the ‘CIO Workshop 1: Basic Investment Concepts’. However, ‘Investigate how agencies protect investors and regulate financial markets and products’ (Standard 3) was not found to be fully covered by the IIG curriculum. Students learned about the fundamentals of investments and the stock market, however it was unclear whether the IIG curriculum provided information on how agencies protect investors and regulate financial markets and products.

‘Risk Management and Insurance’ was another category that showed alignment across all standards. ‘Identify common types of risks and basic risk management methods’ (Standards 1) was aligned with the information covered in the ‘CEO Workshop 2: Income tax and Insurance’. ‘Justify reasons to use property and liability insurance’ (Standard 2) was also consistent with the same workshop, ‘CEO Workshop 2: Income tax and Insurance’. ‘Justify reasons to use health, disability, long-term care, and life insurance’ (Standard 3) was aligned with the ‘CEO Workshop 2: Income tax and Insurance’.

Lastly, in the ‘Financial Decision Making’ category, ‘Recognize the responsibilities associated with personal financial decisions’ (Standard 1) was aligned with the contents covered in the ‘CFO Workshop 2: Money Management’, as well as the ‘CEO Workshop 1: Student Loans’. ‘Use reliable resources when making financial decisions’ (Standard 2) overlapped with the information provided at the ‘CFO Workshop 2: Money Management’. ‘Summarize major consumer protection laws’ (Standard 3) was not aligned with the IIG curriculum, however, the CFO workshops might be a good place to add this topic. ‘Making criterion-based financial decisions by systematically considering alternatives and consequences’ (Standard 4) was covered during the ‘CFO Workshop 2: Money Management’. ‘Apply communication strategies when discussing financial issues’ (Standard 5) was found partially covered by the ‘CFO Workshop 2’, however this is a topic that can be covered across different workshops. One appropriate place to include this is the ‘CFO Workshop 2: Basic Money Management & Creating a Spending Plan’. ‘Analyze the requirements of contractual obligations’ (Standard 6) did not seem to be aligned with the IIG curriculum. However, this can be applicable to several workshops, when students are learning about employment (Workshop 3), insurance (Workshop 2), and/or possibly credit card agreements in the CFO section (Workshop 4). ‘Control personal information’ (Standard 7) was not found to be covered by the IIG curriculum, however this is believed to be a fit for the CFO section Workshop 3 or 4, when students learn about financial accounts and credit cards. Lastly, ‘Use a personal financial plan’ (Standard 8) was discussed throughout multiple IIG workshops, including ‘CFO Workshop 2: Money Management’, ‘CIO Workshop 3: Retirement Saving’, and ‘CEO Workshop 1: Student Loans’.

The comparisons between the national standards and the IIG curriculum provided an overview of what consists of the IIG workshop modules and their alignment with the national effort in making financial education consistent throughout the nation. The areas that were found to be misaligned will be discussed later in relation to the student interview responses and areas for improvement.

## 3. Methods

The Principal Investigator and her research team conducted a mixed-methods quasi-experimental separate-samples pretest-posttest study to assess the efficacy of the IIG program and have been conducting a longitudinal qualitative study to examine the impact of the program on participants’ knowledge, behavior, and future decisions. As the IIG program was delivered as an elective afterschool program, all girls who participated in the program were invited to be part of the study. The workshops were offered in person and/or remotely by IIG staff who were trained with its curriculum. The first study evaluated the lesson quality based on the workshops conducted during the 2018–2019 academic year. The second study assessed the impact of IIG workshops on the girls based on the one-on-one interviews conducted with IIG alumnae, the 2019–2020 student cohort. While the longitudinal study continues until 2023, this paper reports on the preliminary results, focusing on the first-year longitudinal survey, and it should be noted that the findings are not conclusive yet. More data collection and analysis are underway, and it is expected to show more concrete evidence on girls’ long-term life decisions in the coming years.

The separate-samples pretest-posttest strategy is designed to evaluate whether exposure to the financial literacy curriculum improved self-efficacy (quantitative items), understanding of financial concepts, and motivation to engage in financial literacy (qualitative items). The research strategy was implemented in the following manner:As the youths entered the classroom, the project director randomly provided each student with one of two color-coded folders containing the evaluation protocols;One packet asked that the youths complete the quantitative survey items prior to beginning the workshop and subsequently requested they respond to the open-ended item after completing the workshop;The other packet requested that the youths complete the open-ended item prior to beginning the workshop and subsequently asked them to respond to the quantitative items after completing the workshop.

The separate samples pretest-posttest design strategy is robust with respect to being able to conclude that changes in responses are likely due to the quality of the financial literacy curriculum. By focusing on conducting a pretest and posttest within each classroom workshop, the design strategy was flexible and responsive to the varied conditions at each school. These conditions included changes in timing for when the workshops were conducted, diversity with respect to whether workshops were conducted in a one-hour or multi-class period arrangement, economic differences between schools, an inability on the part of the project directors to know which youths would be present at a given workshop, and periodic requests by schools to combine the content of two or more workshops into a unique implementation configuration.

IIG offers a unique set of challenges for program evaluation. Each workshop consisted of different participants, which meant that the effect of each workshop needed to be assessed independently from one another. Therefore, it was not possible to track individual participants across two or more workshops, in which they may have been part of. In response to these unique circumstances, a quasi-experimental strategy that experimentally assigns when students are asked to complete a pretest or posttest for each workshop was deemed appropriate. By comparing the mean pretest and posttest scores, the evaluation strategy assessed whether IIG was meeting its goals with respect to increasing girls’ financial literacy.

The longitudinal study used a semi-structured interview protocol where each interview starts with the predetermined questions but may not necessarily follow the list as a new discussion theme is raised by the participant. Each participant is interviewed once a year via Zoom for approximately 30 min. The interviews typically begin by reviewing the IRB consent information, followed by Q&A, conducting the actual interview, and asking for final comments with and without recording.

### 3.1. Measure

Self-efficacy items were designed to address the objectives for each workshop. Project directors identified between three and seven financial literacy objectives that would be addressed in each workshop. Each financial literacy objective was written in a manner that is consistent with self-efficacy measurement using the opening stem “How confident are you that you could…” and with the financial literacy skills from the specific workshop being used as the items that were rated using a five-point (1–5) scale. Sample questions included: “How confident are you that you could identify your money personality?”, “How confident are you that you could explain the importance of saving money?”, and “How confident are you that you could list the education and certifications required for certain finance careers?”

For the qualitative items, participants were asked to describe critical financial terms, why they are important and how they perceive these concepts, using one open-ended item per workshop. Sample questions include: “Why is it important to have a savings plan?”, “What does it mean to use a credit card wisely?”, and “What do you know about the types of student loans available to pay for college?”

The interview questions focused on the impact of the IIG curriculum on students’ financial knowledge, attitudes, and behavior, as well as their plan for postsecondary education and their career. Sample questions included: “When you think back to the IIG curriculum what are the topics that you found to be most helpful?”, “How have these [financial literacy] skills supported your transition into adulthood?” “What are your current career and life goals?”, and “How has your experience in learning about financial literacy affected your decision about these goals?”

### 3.2. Participants

The separate samples pretest-posttest study is the second iteration of the evaluation conducted in 2017–2018. Compared to the previous year, the number of participants in 2018–2019 tripled, from 308 to 1010 female youths. The participants were from 12 schools, with 4 being private schools and the rest being public or public charter schools. Seven schools are considered to be located in the community with “some assets” according to the Opportunity Index [41]. Three schools are considered “average”, and two are believed to have “challenges”. They are located in the US Northeast region, including New York, Connecticut, Maryland, and Massachusetts. The number of workshop topics examined also increased from 4 to 12 topics in the 2018/19 study.

The first-year cohort who participated in the longitudinal interviews included 15 female students. The students who were interviewed in fall 2019 were college freshmen who graduated from high school in May 2019, and the ones interviewed in spring 2020 were high school seniors who had participated in the IIG program in the past. While the pretest-posttest survey participants were asked to be part of the study before and after each workshop session, the interviewees were recruited through the IIG directors, and their contact information was forwarded to the research team.

### 3.3. Plan for Analysis

The self-efficacy items were analyzed using a one-way analysis of variance to compare the pretest and posttest results. The interview data were transcribed and coded, using the qualitative data analysis software NVivo. Thematic analysis was conducted with the interview data and the open-ended items of the separate sample pretest-posttest. We combined the thematic analysis steps by Braun and Clarke [42] with the hybrid coding process of qualitative data that starts deductively and then creates new codes inductively as they emerge [43,44]. The research team: (a) analyzed a few sample responses to be familiarized with the data; (b) identified the responses that have meanings; (c) identified predetermined codes based on the survey questions asked; (d) modified the code list, added new codes that are emerging, creating an initial set of codes; (e) expanded the analysis to a larger number of responses; and (f) refined codes and merged them into themes if appropriate. Major themes will be discussed, as well as some meaningful themes that are categorized under “IIG experience” that manifest the impact of the program on their postsecondary education and career.

## 4. Results

### 4.1. Self-Efficacy Items

Overall, the participants reported significantly higher confidence (self-efficacy) for engaging in financial literacy, for topics in which they were part of. Pretest and posttest responses were received for 1010 girls who participated in CFO, CIO, or CEO workshops between fall 2018 and spring 2019. For self-efficacy, participants reported significantly more confidence in being able to use the financial literacy skills they learned in their workshops (posttest average = 3.80, SD = 0.855 [*n* = 438 students; 95% CI = 3.71–3.89]; pretest average = 2.87, SD = 1.015 [*n* = 572 students; 95% CI = 2.80–2.95]; [F (1, 1008) = 236.24, *p* < 0.000, η2 = 0.19]). According to Cohen [45], an η2 (partial eta2) of 0.19 can be considered a meaningful, large effect size. More descriptive statistics by each school are provided in Appendix A.

### 4.2. Open-Ended Items

The participants became more articulate in describing different financial concepts after the workshop. They were able to provide more details, and below are some sample responses from CIO Workshop 1 and CEO Workshop 3.

CIO Workshop 1 (Becoming the CIO of your life 1: Basics of Investing & Stock Market):

Comparing qualitative responses, it was found that the workshop enabled participants to become more articulate in describing different types of investments and identifying potential issues to consider in the investments:Pre-Workshop sample response: “[When thinking of investing,] savings, a house, and a car come to mind”;Post-Workshop sample response: “Bonds, the amount of interest you need to give back. Stocks, how risky is the stock, mutual funds, the group and what the group does, [come to mind when thinking of investing]”.

CEO Workshop 3 (Becoming the CEO of your life 3: How to Get the Job You Really Want: What makes a strong resume, how to decide between multiple job offers, and the importance of salary negotiations):

Comparing qualitative responses, it was found that the workshop helped participants consider more than salaries when deciding between job offers:Pre-Workshop sample response: “Pick the one with more money”;Post-Workshop sample response: “See how they can help your skills and look at work culture”.

### 4.3. One-on-One Interviews

There were four major themes identified from the fifteen interview transcripts that had several sub-themes. The major categories included: (1) most helpful curriculum topics (49 references); (2) financial literacy skills they are expecting to use next year (20 references); (3) the financial literacy skills that helped them transition from high school to college (14 references); and (4) additional curriculum areas they would like to learn more about (13 references). There were some additional themes that had the potential of connecting to long-term life decisions. Some of these sample responses are provided at the end to share the students’ experiences with Invest in Girls. While these quotes showed the program’s impact on their career and life decisions, notably, the paper covers the first year of the longitudinal survey only.

#### 4.3.1. Most Helpful Topics

There were thirteen sub-themes identified under this category. The two most helpful financial education topics were ‘Budgeting and Credit Cards’, both identified by eight participants, respectively. ‘Retirement’ was mentioned by six participants, followed by ‘College Finance’ (five references), ‘Savings’ (four references), and ‘Stocks’ (four references). ‘Stock’ was a topic that was not discussed by the student cohort interviewed in fall 2019. ‘Career in Finance’, ‘Loan and Debt’, and ‘Salary Negotiation’ were the topics mentioned by three participants, respectively. Two participants commented on ‘Reading Sheets and Balance’. Other helpful topics identified included: ‘IIG Senior Project’, ‘Managing Finance’, and ‘Taxes’.

#### 4.3.2. Financial Literacy Skills Expected to Use Next Year

‘Budgeting’ was most frequently mentioned as the skill that IIG alumnae were expecting to use in the next year (10 references). This is consistent with the results reported above; those participants found the topic of ‘Budgeting’ most helpful. ‘College Finance’ was a theme identified by two participants and a new theme that was not discussed among the 2019 cohort. Other themes commented by the participants included: ‘Accounting’, ‘Investing’, ‘Passing on Financial Skills to Friends and Family’, ‘Saving’, and ‘None’, which was identified by one participant, respectively. Some new themes mentioned by the spring 2020 cohort included: ‘Credit Cards’; ‘Increased Awareness’; and ‘Taxes’ (identified by one participant, respectively).

#### 4.3.3. Financial Literacy Skills That Helped Them Transition to College

When asked about the skills that helped/would help them transition from high school to college, three of the participants described making ‘Wise Spending Decisions.’ Another three mentioned ‘Wise Use of Credit Cards’, and another three identified ‘Financial Independence’ as the skills that helped/would help them transition to college. Two participants mentioned ‘Foundational Skills’ as the valuable financial literacy skills for transition, which was not a theme identified among the 2019 cohort. Other themes identified included: ‘Saving for College’; ‘Scholarship Process’; and ‘Emerging Planning’ (identified by one participant, respectively).

#### 4.3.4. Areas of Additional Financial Literacy Curricula

‘Stock and Investing’ and ‘Taxes’ were the two topics that the students were hoping to learn more about. They were either not introduced to the students or provided insufficient detail and, as such, the students wanted to know more if there was a chance. Two participants wanted to learn more about ‘Credit Cards’. Other topics the participants were interested in learning more about included: ‘Employment Process’; ‘Home and Life Insurance’; ‘Networking’; ‘Retirement Plans’; and ‘Trips to Banks’ (identified by one participant, respectively).

#### 4.3.5. Experiences with Invest in Girls

Many participants described how IIG impacted their career decisions and lives in general. Some of the meaningful conversations are included below.
*“I kind of wanted to be a political science major in my freshman year. I think [IIG] did help me understand why I actually like working with numbers like the money side of things. So, I think it did play a big role, figuring out what I really wanted to do.” (Student M)*
*“IIG is really beneficial for me, just like in my life. Like I am gonna have to be able to manage my own money and go defend myself in front of a future boss who wants to pay me less. And I feel like that was really, really helpful overall.” (Student B)*
*“I feel like a lot of jobs in the financial industry are kind of a boys club… I got a lot of, like valuable information about advocating for myself in a boys club.” (Student A)*
*“A lot of the times, I do feel as though like girls won’t speak up just because they are nervous about boys looking at… Sometimes I would notice that they’ll be scared to say what they feel or ask the question because they don’t want to be looked like they’re stupid. And it’s like [IIG] made a really comfortable space. And you build relationships with your mentor; you mentor other girls around you who are learning just as much as you are willing to and not call you stupid for not knowing, because not everyone knows about financial literacy. And yes, I think that’s the really important part.” (Student C)*


While conversations on improvement areas will be discussed further in the next section, with a focus on the curriculum components, below describes some of the concerns the participants had regarding the program structure.
*“I wish the forum was a little bit bigger. This year I think everyone got into the program. But I think maybe the year before, a couple of years ago, I know they didn’t have enough space. They had to deny a few people. I think the whole school should take it honestly.” (Student MD)*
*“I wish this program started earlier like high school freshman or high school sophomore, kind of exploratory opportunity of some sorts.” (Student D)*
*“I wish we would have had like the same teacher throughout all three years. I think a lot of my cohorts did have that, but we did have a switch. So, it’s a little bit interesting trying to get to know a new teacher while learning new material. She had a very different teaching style than our old teacher, but we connected very well. That switch was a little bit difficult in terms of learning the material.” (Student S)*
One of the participants summarized the value of the program in addressing the inequity issue: *“I would love to still be a part of Invest in Girls, but sadly, I can’t. I’m just hoping that you guys keep this program alive and teach other young women—any color, any race, about being a businesswoman. Please put more programs in cities or more and more places. There are kids out there that really want to learn. It would be very helpful for them.”*


## 5. Discussion

The present study contributes to the existing literature by providing evidence on how a financial literacy intervention has a positive impact on youth knowledge, behavior, and autonomy development, which are considered part of career planning and management skills that prepare them for the future. The results from the self-efficacy items indicated that the participants reported significantly higher confidence for engaging in financial literacy, as well as a higher ability to articulate financial concepts after the IIG workshop. The findings from the one-on-one interviews also informed us that the program was influencing students in positive ways, enabling them to build financial literacy skills, such as budgeting and wise use of credit cards, and reconsider their future pathways. Consistent with the existing literature, the program had a positive impact on the students’ knowledge around complex topics (e.g., stocks, taxes) [13,46] and their financial behavior (e.g., planning for college, emergency, and retirement) [15,16,17] and autonomy development (e.g., financial independence) [24]. It is meaningful in that the study strengthens the existing literature that focuses on the female population, such as increased saving behavior among participating girls after learning about saving [19] and improved financial autonomy among homemakers [24]. This is also consistent with the findings of Luhrmann et al. [9], in that the intervention raised participants’ interest in learning more about finance. The present study moves one step further by providing evidence that the increased interest in finance often led to girls’ decision to pursue educational opportunities relevant to finance.

The findings are important in that they focused on the female population and how the intervention impacted their transition from high school to college, which are the two research areas currently under-explored in the literature. For their transition, the information and skills they were able to build in relation to making wise spending decisions were helpful, in particular. The first-year longitudinal study results also informed us that the program has the potential to address the inequities within the world of finance and impact the future goals and long-term life decisions of female students. The research team will continue to follow-up with the IIG graduates to see the long-term impact of the program. While the longitudinal study continues until 2023, this paper reported on the preliminary results, focusing on the first-year longitudinal survey, and the findings are not conclusive yet.

### Implications for Practice

Students also suggested the areas that have the potential for improvement. In examining these with our comparison between the IIG curriculum and the National Standards, there are four major suggestions to be considered by those designing a financial literacy program: (a) ‘Budgeting’ is an essential financial education topic that students identify as the most helpful and the skill they are expecting to use in the near future. The recommendation is that the lessons around budgeting are designed practically with a variety of tools and resources, and with the intent of immediate use; (b) We identified that some students were interested in learning more about credit cards. We recommend that the interventions consider in-depth discussions of credit, and for the IIG program to consider adding information on major consumer credit laws to be aligned with the national standards; (c) ‘Stock and investing’ was another topic that the students were interested in learning more about. We recommend that the interventions consider introducing more depth into the curriculum, such as investing as a minor, opening a custodial account, and the ways to begin investing as they turn 18. The IIG program can align its curriculum to the national standards by adding information on how agencies protect investors and regulate financial markets and products; (d) ‘Understanding the employment process’ is also essential considering the transition from youth to adulthood. Financial education interventions are advised to include a lesson on how to read and understand a contract agreement before entering such an agreement. To align with the national standards, the recommendation is that IIG add an activity on how to analyze the requirements of contractual obligations.

More suggestions that are particular to the IIG program emerged during the interviews. ‘Stock/Investing’ and ‘Taxes’ were the two topics most frequently mentioned by the students as topics to learn more about. The recommendation is that the IIG program examine whether the lessons on these two topics cover enough information and consider including in-depth content. Participants also mentioned ‘Credit Cards’ as an area to explore more. Other topics the participants were interested in learning more about included: ‘Employment Process’; ‘Home and Life Insurance’; ‘Networking’; ‘Retirement Plans’; and ‘Trips to Banks’ (identified by one participant, respectively). In addition to the suggestions on the IIG curriculum, the participants shared their concerns about the program structures. It is recommended that the IIG program work with the participating schools to: (a) maximize the number of students who can participate; (b) begin the program in the first or second year of high school if possible; and (c) identify reasons for teacher turnover to address the issue. While these are specific to the IIG program, other financial literacy programs can consider them in designing the programs.

An area of research that is not discussed in this study but is critical in understanding the financial knowledge and behavior of youth is the involvement of the family. The interview participants in this study often mentioned the impact of their family members on their spending and saving decisions. They described shaping their habits of wise money use from either information gained in conversations with older family members or simply imitating what their family members do. Female family members can play a significant role in narrowing the gender gap in financial literacy, and we need further investigation on how such informal relationships have a short and long-term impact on female students and their transition to a postsecondary education and to their career.

Lastly, it is arguable whether the gender disparity in the leadership roles in finance is an issue that can be addressed through education initiatives or a bigger issue that requires changes in the education system as a whole. Our belief is that while changing the whole system can address many related problems, changing a system takes time, and we often need to provide evidence through programs like IIG to convince policymakers and support practitioners. As a translational research effort [47], this intervention translates research to practice, examines the benefits and effectiveness of the IIG program, and seeks to move the intervention, if found effective in the long run, into large-scale practice.

## 6. Conclusions

The Invest in Girls program aims to increase the number of girls pursuing the finance major and of female leaders in the finance field. When girls see more female executives and leaders in finance, they are more likely to consider the field as an accessible and viable career pathway. Currently, we provide few role models for girls, with less than one-quarter of the leadership positions filled by women in the United States [8]. As part of an effort to help address the gender disparity issue, the purpose of this study was to examine the effectiveness of the IIG program that promotes girls’ financial literacy skills and career planning and management skills, which ultimately increase their college and career readiness. The findings of this research showed that it is possible to change the trend of gender disparity by helping female students build financial knowledge and habits of wise spending and decisions and providing more quality educational opportunities that would help them consider pursuing their future in finance. Our hope is that more financial education programs like Invest in Girls, that focus on girls, become available to our underserved populations and contribute to empowering and helping them become “future-ready”.

## Data Availability

Data is contained within the article.

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
