# Peer review of "Empowering Women in Finance through Developing Girls’ Financial Literacy Skills in the United States"

_behavsci, 2021, doi:10.3390/bs11120176_

Round 1
Reviewer 1 Report
Review: behavsci-1409141
Thank you for the opportunity to review the manuscript, “Empowering Women in Finance Through Developing Girls’ Financial Literacy Skills in the United States”. The focus on promoting financial literacy among females is an important topic. The use of a multi-method approach to assess program impact is also a strength. Despite these strengths, it was difficult for me to follow the narrative, and connect the narrative to the results. In contrast to the introduction and lite review, the discussion is underdeveloped, and leaves the reader questioning what new insights and knowledge have been revealed. Finally, further details about the method would be helpful in understanding who the participants are - and when they participated. In summary, the study has potential, but the manuscript needs to be restructured to make it clear what the study did - and what has been found. Here are my main suggestions for revising the manuscript:
- The introduction and review of the literature is quite long, and covers perhaps too broad a range of topics. It might be helpful to provide a conceptual model describing the hypothesized associations between the program and the results (e.g., the associations between components of IIG and expected outcomes). Then streamline the evidence presented to support the components and associations. What is the importance of introducing state policies for this specific program/study? Similarly, do the students in this program develop ILP’s? As for the mapping to the National standards, could this perhaps be better handled as a table or matrix, perhaps in an Appendix? In other words, the reader has to figure out how all this information relates to the present study, and I am not convinced it is needed.
- Relatedly, the authors introduce a variety of financial education terms, but how these various terms related in the context of the present study? What is meant by financial capacity? Is this the same as financial capability (FINRA, 2015; Johnson & Sherraden, 2007; Serido et al., 2013)? The explanation regarding the association between financial literacy and financial education is confusing. Can you provide a simpler definition? Much of the literature defines financial education as teaching of financial content as a way to improve financial knowledge and financial literacy as the ability to apply learned knowledge to enact new behavior.
- Could you provide additional information about the study design (after school/during school classes; mandatory/elective; more information about the participants; educator training etc.. demographics)
- It would seem that the unique aspect of the study is the focus on females across sociodemographic. Can you elaborate on this more? Perhaps the authors could move/incorporate some of the evidence/findings in the literature into the discussion to provide a more robust understanding of the value of the present study.
Author Response
Point 1: The introduction and review of the literature is quite long, and covers perhaps too broad a range of topics. (a) It might be helpful to provide a conceptual model describing the hypothesized associations between the program and the results (e.g., the associations between components of IIG and expected outcomes). Then streamline the evidence presented to support the components and associations. (b-1) What is the importance of introducing state policies for this specific program/study? (b-2) Similarly, do the students in this program develop ILP’s? (c) As for the mapping to the National standards, could this perhaps be better handled as a table or matrix, perhaps in an Appendix? (d) In other words, the reader has to figure out how all this information relates to the present study, and I am not convinced it is needed.
Response 1: (a) We introduced ILP as a conceptual model that enables youth to develop financial knowledge and behaviors as part of their career planning and management skills. With respect to the association between the program and the expected outcomes, we clarified that the purpose of this paper is to examine the impact of the IIG program on girls’ knowledge and behaviors, as opposed to what the IIG program ultimately aims to – address the gender disparities existing in the finance industry. (b) We provided information on the state policies and ILP to highlight how financial education has been promoted at the state level and helped shape the IIG curriculum - The increasing number of states adopting financial education resulted in the establishment of the national standards, which provide a framework for IIG and help maintain consistency across different financial literacy programs. (c) For a better visual representation, we created a matrix table that compares the National Standards to IIG (see Appendix). (d) Throughout the manuscript, we reiterated the purpose of this study and differentiated it from the broader purpose of the IIG program.
Point 2: (a) Relatedly, the authors introduce a variety of financial education terms, but how these various terms related in the context of the present study? (b) What is meant by financial capacity? Is this the same as financial capability (FINRA, 2015; Johnson & Sherraden, 2007; Serido et al., 2013)? (c) The explanation regarding the association between financial literacy and financial education is confusing. Can you provide a simpler definition? Much of the literature defines financial education as teaching of financial content as a way to improve financial knowledge and financial literacy as the ability to apply learned knowledge to enact new behavior.
Response 2: (a) Clarified how the various terms are connected to the present study in the Definition section. The paper describes various definitions of financial literacy, from narrow to broad definitions that extends to our social responsibility, to highlight the roles of financial education in addressing different societal issues, including the gender wage and career advancement gaps. (b) The definition of financial capacity has also been added, and we have changed financial capacity to financial capability to align ourselves with the existing literature. (c) We also provided simpler definitions for financial literacy and financial education as suggested.
Point 3: Could you provide additional information about the study design (after school/during school classes; mandatory/elective; more information about the participants; educator training etc.. demographics)
Response 3: Provided additional information about the study design, including whether or not it was an after-school program and an elective course. We added that the workshops were provided by IIG staff who are trained with the IIG curriculum and financial literacy, in general.
Point 4: It would seem that the unique aspect of the study is the focus on females across sociodemographic. Can you elaborate on this more? Perhaps the authors could move/incorporate some of the evidence/findings in the literature into the discussion to provide a more robust understanding of the value of the present study.
Response 4: Incorporated some of the findings in the literature that are related to the female population into the Discussion.
Reviewer 2 Report
The paper's starts with a title that states a hypothesis: that low rates of women participation in entrepreneurial area and finance could be addressed through developing girls' financial literacy skills.
This hypothesis is not easy to be confirmed in the short-run and although the paper conclusion tries to assure that the IIG program had a positive effect in girls' long-term life decision, the results do not seem to confirm such a conclusion.
With regard to empirical data that were collected after a series of 3 workshops (but only after each singe workshop) it is stated that the study compares the mean pretest and posttest scores, although it does not provide any other statistical measures such as standard deviation, or if the distribution is symmetrical or not.
However, regardless of the descriptive statistics that are not analytically provided, the final result (that education increases girls' financial literacy) was expected, even without any statistical analysis.
Nevertheless, this result is not adequate to support that after one, two, or three workshops the percentage of the girls that will follow an financial career would be greater, than a hypothetical percentage of a sample that would not get this type of education. But even if this was the case, it has to be tested and confirmed in the long-run.
Moreover, the second longitudinal study (the one with the interviews) although it was expected (according to the abstract) to show a positive impact in girls' long-term life decisions, the analysis and the results do not mention such a conclusion (or it is not obvious in the text).
Also, in the section "Experiences with Invest in Girls" all the participants are happy and thus this section seems to promote IIG and not to function as feedback in order to improve something or refine the educational procedure.
In other words, although the paper analyses -very well- a well-known problem regarding the low women participation in financial sector, the technical analysis and the results either were expected (first case) or are not confirmed (second study).
After reading this paper the question remains. Could the issue of low-rates of women participation in financial sector and especially in managerial positions as business leaders be addressed through educational initiatives or the educational procedures as a whole should be transformed.
Author Response
Point 1: The paper's starts with a title that states a hypothesis: that low rates of women participation in entrepreneurial area and finance could be addressed through developing girls' financial literacy skills.
This hypothesis is not easy to be confirmed in the short-run and although the paper conclusion tries to assure that the IIG program had a positive effect in girls' long-term life decision, the results do not seem to confirm such a conclusion.
Response 1: Clarified what the purpose of the IIG program is – addressing the gender disparities existing in the finance industry – and what the purpose of this paper is – examining the impact of the IIG program on girls’ knowledge and behaviors and their decision to pursue educational opportunities leading to finance careers. We also added more quotes from our longitudinal study that show how their experiences with the program influenced their interests and provided a sense of comfort in pursuing a career that has historically been male-dominant.
Point 2: With regard to empirical data that were collected after a series of 3 workshops (but only after each singe workshop) it is stated that the study compares the mean pretest and posttest scores, although it does not provide any other statistical measures such as standard deviation, or if the distribution is symmetrical or not.
However, regardless of the descriptive statistics that are not analytically provided, the final result (that education increases girls' financial literacy) was expected, even without any statistical analysis.
Nevertheless, this result is not adequate to support that after one, two, or three workshops the percentage of the girls that will follow an financial career would be greater, than a hypothetical percentage of a sample that would not get this type of education. But even if this was the case, it has to be tested and confirmed in the long-run.
Response 2: In addition to clarifying the purpose of this study (c.f., the aim of the IIG program), we provided additional descriptive statistical by each school in Appendix. We agree that it has to be examined in the long run and will follow up with the IIG graduates until the first interview cohort graduates from college.
Point 3: Moreover, the second longitudinal study (the one with the interviews) although it was expected (according to the abstract) to show a positive impact in girls' long-term life decisions, the analysis and the results do not mention such a conclusion (or it is not obvious in the text).
Response 3: This paper includes preliminary findings of the longitudinal study. We edited the abstract to reflect that additional data collection and analysis are underway and that the findings cannot be conclusive until the end of the longitudinal study.
Point 4: Also, in the section "Experiences with Invest in Girls" all the participants are happy and thus this section seems to promote IIG and not to function as feedback in order to improve something or refine the educational procedure.
Response 4: Added more feedback received from the individual participants. Implications for Practice begins with a list of recommendations for program improvement, drawn from two sources: a) comparisons between the national standards and the IIG program and b) the feedback received from the individual participants.
Point 5: In other words, although the paper analyses -very well- a well-known problem regarding the low women participation in financial sector, the technical analysis and the results either were expected (first case) or are not confirmed (second study).
After reading this paper the question remains. Could the issue of low-rates of women participation in financial sector and especially in managerial positions as business leaders be addressed through educational initiatives or the educational procedures as a whole should be transformed.
Response 5: We clarified the purpose of this study (c.f., the aim of the IIG program) and provided additional information on the status of the longitudinal study. We agree with the reviewer’s comment on the systems issue. The whole system change is certainly a strategy that can address many related issues, but it is our point of view that changing a system takes time and we often need to provide evidence through programs like IIG to convince policymakers and support practitioners so that they could adopt and implement quality programs. We provided information on Translation Research (Center for Disease Control and Prevention, 2018) as we are seeing the IIG program as part of an effort to translate research to practice and, ultimately, into large-scale practice if found successful.
Round 2
Reviewer 1 Report
Thank you for investing in the clarity and flow of the narrative. I appreciate the including of additional detail on the method. I have no further comments or concerns.
Author Response
Response to Reviewer 1 Comments
Point 1: Thank you for investing in the clarity and flow of the narrative. I appreciate the including of additional detail on the method. I have no further comments or concerns.
Response 1: Thank you very much. We appreciate your time and effort to review the revisions.
Reviewer 2 Report
Any initiative that aims in reducing inequalities that is consistent with UNESCO's SDGs is wellcomed. More than welcomed are the initiatives that present their way of working and researching the effects of their work.
The authors have tried to get back with a revised version of their paper. In my point of view, this is an improved version but still there is one point that has to be improved.
The section "Experiences with Invest in Girls" still looks like advertising IIG. Nothing is mentioned about feedback or about some design insights that come from compiling the experiences with IIG. What is the role/aim of such a section if no actions were taken based on those experiences. Are all the experiences that positive? If the authors are really interested in improving the efficiency and effectiveness of their effort they should respond with empathy to any participants' experience good or bad and try to provide an improved design of their initiative.
Author Response
Response to Reviewer 2 Comments
Point 1: Any initiative that aims in reducing inequalities that is consistent with UNESCO's SDGs is wellcomed. More than welcomed are the initiatives that present their way of working and researching the effects of their work.
The authors have tried to get back with a revised version of their paper. In my point of view, this is an improved version but still there is one point that has to be improved.
The section "Experiences with Invest in Girls" still looks like advertising IIG. Nothing is mentioned about feedback or about some design insights that come from compiling the experiences with IIG. What is the role/aim of such a section if no actions were taken based on those experiences. Are all the experiences that positive? If the authors are really interested in improving the efficiency and effectiveness of their effort they should respond with empathy to any participants' experience good or bad and try to provide an improved design of their initiative.
Response 1: Thank you for your comments. We have added some of the students’ concerns regarding the program structure (e.g., workshop length, timing, instructor turnover; 701-716; 788-794). Their comments related to the IIG curriculum are discussed in Implications for Practice as areas for improvement.